# Universe of Thoughts: Enabling Creative Reasoning with Large Language Models

## Abstract

Reasoning based on Large Language Models (LLMs) has garnered increasing attention due to outstanding performance of these models in mathematical and complex logical tasks. Beginning with the Chain-of-Thought (CoT) prompting technique, numerous reasoning methods have emerged that decompose problems into smaller, sequential steps (or thoughts). However, existing reasoning models focus on conventional problem-solving and do not necessarily generate creative solutions by "creative reasoning". In domains where the solution space is expansive and conventional solutions are suboptimal, such as drug discovery or business strategization, creative reasoning to discover innovative solutions is crucial. To address this gap, first we introduce a computational framework for creative reasoning inspired by established cognitive science principles. With this framework, we propose three core creative reasoning paradigms, namely, *combinational*, *exploratory*, and *transformative* reasoning, where each offers specific directions for systematic exploration of the universe of thoughts to generate creative solutions. Next, to materialize this framework using LLMs, we introduce the *Universe of Thoughts* (or *UoT*, for short), a novel set of methods to implement the aforementioned three creative processes. Finally, we introduce three novel tasks that necessitate creative problem-solving, along with an evaluation benchmark to assess creativity from three orthogonal perspectives: feasibility as constraint, and utility and novelty as metrics. With a comparative analysis against the state-of-the-art (SOTA) reasoning techniques as well as representative commercial models with reasoning capability, we show that UoT demonstrates superior performance in creative reasoning. This work introduces a new perspective on how LLMs can become autonomously creative, advancing the field to address problems that require more innovative solutions.

## 1 Introduction

Large Language Models (LLMs) have advanced rapidly, demonstrating impressive capabilities in increasingly complex tasks that require reasoning (Achiam et al., 2023; Chowdhery et al., 2023). However, the fundamental mechanism of LLMs—predicting the next token in a sequence—can inherently limit their deliberative reasoning abilities. To address this limitation, techniques such as Chain-of-Thought (CoT) (Wei et al., 2022) and Tree-of-Thoughts (ToT) (Yao et al., 2023) have been developed. These frameworks structure the reasoning process by prompting the model to break down a problem into a series of intermediate steps or connected "thoughts". By doing so, they have yielded significant performance improvements on a variety of reasoning benchmarks.

However, prevailing reasoning frameworks are primarily engineered for well-defined problems that have structured, verifiable solutions within a specific domain of thoughts, such as mathematical equations or logical puzzles. In contrast, many real-world challenges are ill-defined problems, characterized by ambiguous goals with vast, open-ended solution spaces (Simon, 1973). For instance, in business strategizing, success often requires generating not just a solution, but a creative one (Singh et al., 2024). Similarly, in drug discovery, the chemical compound space is immense, and breakthroughs depend on identifying creative candidates beyond known therapeutic agents (Gangwal et al., 2024).

While existing reasoning frameworks like CoT can be used to produce creative outputs, this "creativity" is not an emergent property of the reasoning method itself. Instead, it relies heavily on meticulous human prompt engineering; hence, human creativity rather than method creativity. Without human guidance to direct the creative reasoning process, these reasoning frameworks will reduce to exhaustive search of the corresponding domain of thoughts, which might be feasible for solving well-defined problems, but infeasible for creative reasoning which may require search for solution in the vast universe of thoughts. This reveals a critical gap: currently there is no LLM reasoning framework specifically designed to perform autonomous creative reasoning to handle ill-defined problems.

To address this gap, we introduce a framework for "creative reasoning" with LLMs. Toward this end, we draw inspiration from cognitive science, specifically from Margaret Boden's seminal work defining three distinct types of creativity: combinational, exploratory, and transformational (Boden, 2004; 2007; 2009). Combinational creativity emerges from generating novel combinations of existing ideas (i.e., thoughts) by searching the universe of thoughts. Exploratory creativity involves discovering new thoughts in the universe of thoughts that can be combined with existing thoughts to generate creative solutions. Finally, transformational creativity, which is considered the most profound form of creative process, involves altering the presumed rules (or constraints) for the solution itself, enabling search for solutions is radically uncharacteristic parts of the universe of thoughts to generate "outside-the-box" solutions.

Furthermore, to materialize our proposed creative reasoning framework, we introduce the *Universe of Thoughts* (or *UoT*, for short). UoT provides the underlying structure to represent, manage, and manipulate the universe of thoughts, and offers three efficient methods to perform combinational, exploratory, and transformative creative reasoning with LLMs in the universe of thoughts, respectively. The proposed UoT methods for combinational, exploratory, and transformative build each other in the same order, reflecting the fact that these creative processes in our framework are also each extension of the previous process in that order.

Finally, given the novelty of the topic of creative reasoning with LLMs, we realize that absence of an evaluation benchmark in the field with tasks that require creative reasoning is a key challenge. To address this limitation, we propose a new evaluation benchmark for creative reasoning methods. With this benchmark, we design three novel, open-ended, and diverse tasks, which require solutions that are not only effective, but also creative. Moreover, following the classic interpretation of creativity from the field of cognitive science (Runco & Jaeger, 2012), where solutions are considered creative if they are both novel and effective, with our proposed evaluation benchmark we use the primary metrics of *Novelty* (which measures the originality of the solution) and *Utility* (which evaluates how well a solution solves the problem) to assess performance of creative reasoning methods. We also evaluate *Feasibility* to ensure the proposed solutions are valid and adhere to the constraints of the problem. Although in our evaluation benchmark we use a powerful LLM as an evaluator (Zheng et al., 2023), we observe that different reasoning methods/models generate solutions at varying levels of granularity, making direct comparison unfair. To mitigate this issue, we introduce a canonicalization step, where an LLM refines and standardizes the structure of all method outputs, ensuring that all solutions are evaluated on a consistent and equitable basis. With extensive evaluation using our evaluation benchmark, we demonstrate that with all three tasks, UoT outperforms the state-of-the-art (SOTA) reasoning techniques as well as some representative commercial models with reasoning capability such as GPT-5.

## 2 RELATED WORK

### 2.1 LLM REASONING FRAMEWORKS

Large Language Models (LLMs) have inspired numerous reasoning frameworks that focus on improving how these models explore thoughts and solution spaces. Chain-of-Thought (CoT) prompting was an early example. CoT improves complex reasoning by eliciting step-by-step intermediate thoughts (Wei et al., 2022). Tree-of-Thoughts (ToT) generalizes CoT to a search tree, allowing the model to consider branching reasoning paths, backtrack, and self-evaluate choices (Yao et al., 2023). More recently, reasoning methods further generalize to search graphs of thoughts (instead of trees of thoughts) in order to expand the combinatorial capability in reasoning. For example, Graph-of-Thoughts (GoT) constructs an arbitrary graph of thoughts where partial solutions can merge and

feed back into each other, combining different LLM thoughts into synergistic outcomes (Besta et al., 2024). Such graph-based prompting has demonstrated gains in tasks like sorting and planning by reusing and refining intermediate results. Other researchers have proposed enhancements to the aforementioned methods (e.g., EGoT), with strategies such as adding self-reflection or rationale aggregation steps to refine the intermediate conclusions of the models, or by dynamically adjusting generation parameters (e.g. temperature) to balance exploration versus exploitation during the search(Shin & Kim, 2025). Another line of work lets LLMs discover task structures or sub-modules autonomously. For instance, Zhou et al. (2024) introduces Self-Discover, which prompts an LLM to compose its own reasoning blueprint out of atomic skills (like step-by-step thinking or critical questioning) before solving the problem. By optimizing the reasoning topology (from chains, trees, and graphs to modular plans), such approaches yield significant gains on hard reasoning benchmarks (including BigBench Hard and math problems) as compared with the previous reasoning methods.

However, all aforementioned reasoning methods are designed to operate largely within a fixed problem domain and confined, familiar solution space (Shin & Kim, 2025). If applied to the "universe" of thoughts (as required for creative reasoning), they fail to scale unless meticulously guided by human, making them nonautonomous creative reasoning methods. These approaches excel at optimizing the reasoning path, but do not provide a mechanism for expanding the solution space itself. On the contrary, UoT is designed to systematically and efficiently generate creative solutions by navigating and connecting disparate knowledge domains.

## 2.2 Creativity in Cognitive Science

Outside of computational sciences, the study of creative reasoning has a rich history in cognitive science. A widely cited framework by Boden distinguishes three fundamental forms of creativity: combinational, exploratory, and transformational (Boden, 2004; 2007; 2009). Combinational creativity refers to generating an unfamiliar combination of familiar ideas (often by juxtaposition or analogy). Exploratory creativity means searching within a fixed conceptual space defined by certain rules or styles to find novel structures that could exist under those rules. Finally, transformational creativity involves altering the defining rules or constraints of a domain, expanding what is even considered possible. These concepts have been influential in cognitive theories of creativity. Classic definitions of creativity also emphasize dual criteria of novelty and value/utility (Runco & Jaeger, 2012) – i.e. a creative solution must be both new and useful for the task. This notion also aligns with evaluation measures introduced in psychology, such as the Torrance Tests (Torrance, 1966) or Amabile's criteria (Amabile, 1982), which require an outcome to be unconventional yet effective to be considered creative.

While some earlier AI models borrow ideas from cognitive science to incorporate creativity in computational systems, they are all domain-specific (e.g., for music generation (Carnovalini & Rodà, 2020)) and they are not formulated as general reasoning algorithms with verifiable outputs. To the best of our knowledge, we are first to explicitly translate combinational, exploratory, transformational creative thinking processes to computational forms for generalized creative reasoning with LLMs. We bridge the LLM reasoning and cognitive creativity strands of research by proposing a computational framework and corresponding methods to perform creative reasoning with LLMs.

## 3 Computational Framework for Creative Reasoning

Margaret Boden's seminal work defines three distinct types of creativity (Boden, 2004; 2007; 2009):

> 1. "**Combinational creativity**, produces unfamiliar combinations [i.e., solutions] of familiar ideas, and it works by making associations between ideas that were previously only indirectly linked."

> 2. "In **exploratory creativity**, the person moves through the [solution] space, exploring it to find out what's there (including previously unvisited locations [i.e., new ideas])— and, in the most interesting cases, to discover both the potential and the limits of the space in question."

> 3. "In **transformational creativity**, a [solution] space or style itself is transformed by altering (or dropping) one or more of its defining dimensions [i.e., rules]."

Inspired by this categorization of the creative process, in this section we propose a computational framework for creative reasoning. First to establish the corresponding terminology for our framework, consistent with existing reasoning frameworks for LLMs, we use the term "thoughts" to represent "ideas" (or concepts); a "solution" is a coherent combination of thoughts devised to solve a problem; and a "solution space" is a set of solutions (among all possible solutions in a problem domain) that are identified and delimited by a set of (implicit or explicit) "rules". Figure 1a shows the universe of thoughts, where each thought is shown as a block (to represent thoughts as building blocks of the solutions). The currently known/discovered problem domains are shown in the universe of thoughts, each with one or more solution spaces, where each solution space consists of a number of known solutions. Some of the thoughts in the universe are already known and included in some solution, and some are not incorporated in any known solution yet (i.e., the thoughts that are depicted out of all known problem domains).

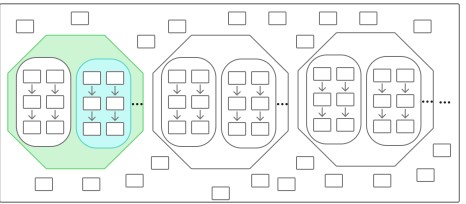

(a) Basic Analytical Reasoning (e.g., CoT, ToT)

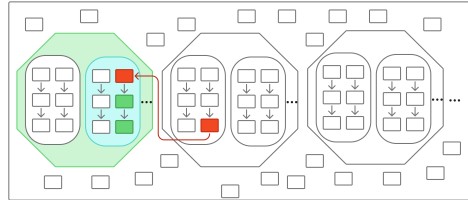
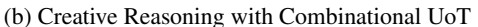

(b) Creative Reasoning with Combinational UoT

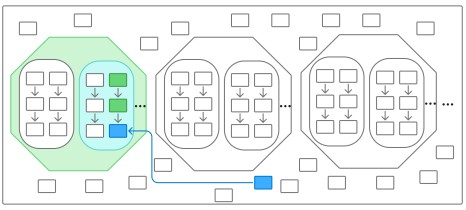

(c) Creative Reasoning with Exploratory UoT

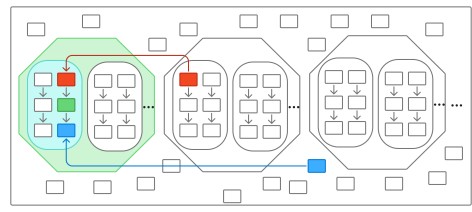

(d) Creative Reasoning with Transformative UoT

Figure 1: *Visualization of basic analytical reasoning methods (a) versus different Universe of Thoughts creative reasoning methods (b-d). While analytical reasoning methods explore thoughts confined to the original solution space (a), C-UoT transfers and combines thoughts from analogous domains (b), E-UoT introduces novel thoughts to expand the existing problem space (c), and T-UoT alters fundamental rules to create a new, transformed solution space (d).*

With this description of the universe of thoughts and its structure, we define our computational framework for creative reasoning. Our proposed framework consists of three computational paradigms to generate creative solutions corresponding to the three creative processes defined in cognitive science, respectively:

1. **Combinational Creative Reasoning**: In this paradigm, one generates novel and creative solutions by (1) identifying known thoughts (i.e., thoughts that have been used as part of some solution in the universe of thoughts) that are relevant to the target solution space of the problem on hand but have not been previously used as part of any solution in this space, and (2) combines these thoughts with existing thoughts in the target solution space. A classic example of combinational creativity is a collage, which combines different types of existing visuals in an unconventional way to create a new piece of art.

2. **Exploratory Creative Reasoning**: This paradigm is similar to the previous paradigm, where the main difference is that the adopted thought(s) from outside of the target solution space are individual thoughts that are not necessarily part of any known solution in the universe of thoughts. In this paradigm, the new thoughts expand the target solution space by introducing new conceptual building blocks that combine with existing thoughts to generate novel solutions. Continuing with our example of painting, when the Impressionism painting style was introduced, creative artists like Monet utilized brushstrokes in a new

functional way, a thought that had not been considered as part of the existing "solutions" to create a painting at that time.

3. **Transformational Creative Reasoning**: This is the most advanced paradigm of creative reasoning, as it involves fundamentally altering or dropping the core rules that define the presumed solution space and provides the opportunity to consider other solution spaces to identify creative solutions for the problem on hand. With this approach the solution space is changed, allowing for discovery of solutions that were previously inconceivable for the target problem. The example here is the Cubism painting style pioneered by Picasso, who broke from the traditional rule of direct representation by depicting objects from multiple angles simultaneously and by fragmenting forms into geometric shapes.

Table 1 shows what is new in each of the creative reasoning paradigms in our proposed computational framework.

Table 1: Novel components in each computational creativity paradigm

| Creativity Paradigm | New Rules | New Thoughts | New Combinations |
|---|---|---|---|
| Combinational | No | No | Yes |
| Exploratory | No | Yes | Yes |
| Transformational | Yes | Yes | Yes |

## 4 Universe of Thoughts: Creative Reasoning with Large Language Models

In this section, we present Universe of Thoughts (UoT), which consists of a set of three methods, each offering a specific implementation/instantiation of one of the three creative reasoning paradigms in our proposed computational framework for creative reasoning, respectively. The complete formalization of the UoT methods is provided in Appendix A. Here, we offer an intuitive description of the underlying mechanism of each UoT method, followed by a running example based on the "Bridge Task" to further illustrate each method. The Bridge Task is one of our proposed benchmark creativity evaluation tasks described in Table 2, where the problem is to design a solution that minimizes average vehicle delay on a single-lane bridge, under the constraint that no new bridges can be constructed.

UoT is implemented as a structured pipeline of modular prompts. Each of the core functions listed below corresponds to a distinct, self-contained call to an LLM, where the description summarizes the specific instructions given to the model:

- **Rule & Assumption Identification:** Identify all explicit and hidden rules governing the target problem domain.
- **Analogous Rule Discovery:** Find functionally similar rules from diverse, analogous domains for each target rule.
- **Rule Set Mutation:** Generate multiple new rule sets by applying mutation operations (e.g., drop, vary, add) using analogous rules.
- **Analogous Problem Discovery:** Identify problems that are structurally similar but different in how they are presented.
- **Analogous Solution Generation:** Find multiple solutions for a given analogous problem.
- **Solution Decomposition:** Decompose all generated solutions into their core conceptual components ("thoughts").
- **Creative Synthesis:** Generate new, hybrid solutions through creative recombination of given thoughts.
- **Evaluation and Ranking:** Assess the generated solutions based on their feasibility, utility, and novelty.

The complete source code and the full prompts used for each function are provided in the supplementary material to ensure reproducibility. Next, we will intuitively describe each of the three UoT methods.

## 4.1 COMBINATIONAL UoT (C-UoT)

Combinational UoT (C-UoT) generates novel solutions by recombining familiar thoughts in unconventional ways. First, C-UoT first identifies analogous problems that are functionally similar to the target problem but perhaps different in appearance. Then it finds a set of known solutions for each analogous problem. These solutions are subsequently decomposed into their constituent thoughts, creating a "pool of thoughts". Next, from the target solution space we select a diverse set of known solutions for the target problem to serve as host solutions. These host solutions are also decomposed into their constituent thoughts to identify their most impactful thoughts. For each impactful thought, we search the pool of thoughts for analogous thoughts that have a similar function but perhaps a different appearance. Finally, we substitute the selected impactful thoughts from the host solutions with their corresponding analogous thoughts to synthesize new candidate solutions, which are then ranked and filtered by feasibility, utility, and novelty.

To elaborate, here we present how C-UoT applies in the context of the Bridge Task as our running example. We first consider analogous single-channel problems—single-track railway dispatch, elevator up-peak control, runway slot assignment, and appointment booking systems. We decompose their solutions into portable thoughts: (i) time-slot reservations / tokens to regulate access, (ii) platooning/batching to cut direction switch losses, (iii) priority windows for critical classes, (iv) ramp-metering at entries, and (v) time-shift incentives to push discretionary trips off peak. Starting from a naïve host policy ("switch direction whenever a car arrives"), we target two high-impact slots—the direction schedule and the arrival interface—and swap in far-but-apt components: a tokenized pre-booking system with 5-minute slots (from runways/clinics), batch service before each switch (from elevators), and off-peak credits for grocery trips (from appointment economics). The synthesized policy still obeys one-lane constraints but reduces switching loss and peak pressure; in our framework this solution is feasible, improves utility (lower delay), and is novel because these components rarely co-occur in standard road-traffic controllers.

## 4.2 EXPLORATORY UoT (E-UoT)

Exploratory UoT (E-UoT) is similar to combinational creative reasoning but it expands the thought palette before recombining the thoughts. To elaborate, alike C-UoT, E-UoT starts by finding analogous problems and decomposing their solutions to build a pool of known thoughts. We then add a single new step to this process: expanding the pool of thoughts with new, functionally equivalent thoughts that were not previously present in any known solutions. To accomplish this, we select a diverse set of anchor thoughts from the pool. For each anchor, we then retrieve or generate candidate thoughts that are functionally similar to the anchor but perhaps different in appearance. The remainder of the process is identical to that of C-UoT.

To continue with our running example, with E-UoT we start with familiar thoughts like slot booking and simple incentives, then add new, functionally equivalent thoughts that look different: e.g., community time-bank credits for off-peak pledges, a group-pledge lottery that rewards blocks for shifting demand, quota tokens that neighborhoods can trade to coordinate semi-peak crossings, and group reservations that auto-cluster bookings into platoons. With this expanded palette, E-UoT then recombines thoughts under the same one-lane rules into a policy like: tokenized bookings plus community-driven incentives with auto-platooning. Such a solution remains feasible (one direction at a time), improves peak spreading and switch-loss reduction, and is novel because it pairs community time-banking with slot reservations—an uncommon mix in road traffic control that still plays the same functional role as standard booking and incentives.

## 4.3 TRANSFORMATIVE UoT (T-UoT)

Transformative UoT (T-UoT) changes the rules that define the target solution space in the target problem domain, then searches for solutions within the newly created solution space. Intuitively, T-UoT first identifies the explicit rules of the target solution space. Then, by analyzing commonalities across known solutions, it also uncovers hidden assumptions and codify them as additional rules. After establishing a comprehensive rule set for the target solution space in the target problem domain, we find analogous rules for each one that are perhaps different in appearance but functionally similar, creating a "pool of rules". From the original rule set, we then identify the most impactful rules. For each of these, we apply mutation operations—either dropping the rule or varying it by substituting an analogous rule from the pool of rules—to generate new rule sets. New rules from the

Table 2: Benchmark Creative Reasoning Tasks

| Task | Description |
|---|---|
| *Task 1: One-Lane Bridge* | *Objective:* Design a policy to minimize average vehicle delay. *Context:* A single-lane bridge with known traffic patterns, fixed crossing times, and direction-switching penalties. *Constraints:* No new infrastructure; traffic must remain unidirectional at all times. |
| *Task 2: Electricity Tariff* | *Objective:* Design a tariff and Demand Response (DR) program to reduce daily peak load. *Context:* A residential feeder experiences high strain during predictable peak hours. *Constraints:* No new physical capacity; power for critical medical loads must never be curtailed. |
| *Task 3: Social Cohesion* | *Objective:* Design an intervention to strengthen cross-group cohesion, measured by new ties, mixing, and mutual aid. *Context:* A community with distinct social groups and a fixed intervention period. *Constraints:* Must protect privacy, be inclusive, and maintain a safe environment. |

pool can also be added to these sets. Finally, at this point we apply E-UoT to the newly established solution spaces for each new rule set to generate novel solutions.

To continue with our example, T-UoT starts by surfacing hidden rules—e.g., people must cross in person, groceries require a shopper trip, only road vehicles are allowed, and access is first-come-first-served (FCFS). The transformational step is to transform these rules: permit supervised non-road corridors (water/air), allow goods to move without people (drone lockers, mobile markets), or replace FCFS with credit allocation. Each transformed rule set allows for solutions that were impossible before (closure escape): drone delivery eliminates many grocery trips; supervised air/water shuttles add a parallel channel; and quota credits reserve scarce peak crossings for essential work.

### 4.4 COMPLEXITY ANALYSIS FOR UOT METHODS

UoT methods all drastically reduce the computational complexity of creative problem-solving as compared to factorial-growth exhaustive search baseline solutions reviewed in Section 2.1. By employing a guided search through analogical domains, UoT prunes the vast search space, resulting in an significant efficiency gain across all three reasoning paradigms. The detailed complexity analysis and formal derivations are provided in Appendix B.

## 5 EXPERIMENTAL EVALUATION

### 5.1 EVALUATION BENCHMARK

#### 5.1.1 TASKS

To evaluate creative reasoning methods, we designed a set of three novel and diverse tasks that require both creativity and logical problem-solving. Each task was specifically constructed with two key properties: 1) an expansive solution space to permit a wide range of potential answers, and 2) a structure that renders conventional or obvious solutions insufficient. Description of these tasks is provided in Table 2.

#### 5.1.2 EVALUATION METRICS

Our evaluation framework is guided by the classic definition of creativity (Runco & Jaeger, 2012), which comprises two primary components: Utility and Novelty. We also introduce Feasibility as

a critical third metric to ensure that solutions are valid and practical by adhering to all problem constraints.

A key challenge in evaluating LLM-generated outputs is their varying granularity. One model might offer a concise core idea, while another provides a detailed plan with sub-solutions. To ensure a fair comparison focused on the generated core creative insight, we introduce a Solution Canonicalization step. Before scoring, each generated solution is processed by an LLM to extract its single, core conceptual idea. This pre-processing step standardizes all outputs to a consistent format, enabling an equitable assessment of creativity. Following canonicalization, each solution is assessed using the metrics below:

- *Utility*: This metric measures how effectively the solution solves the problem. It is scored on a continuous scale from 0 to 1, where a baseline known solution is anchored at 0 and a pre-defined optimal solution is anchored at 1. An LLM evaluator interpolates the score based on the perceived effectiveness of the solution relative to this scale.

- *Novelty*: To accurately measure conceptual novelty, we first create a semantic embedding of the canonicalized core idea. The novelty score is then calculated as its minimum cosine distance from the set of known solutions. To create a meaningful scale, this distance is normalized by the distance between the known solutions and the optimal solution, which serves as a practical upper bound.

- *Feasibility*: This metric is scored from 0 to 1 and assesses whether a solution is plausible and meets the rules of the problem. The LLM evaluator is provided with a pre-defined checklist of all explicit constraints and scores the solution based on adherence to constraints.

The final *Creativity Score* is calculated as the sum of the Utility and Novelty scores for all solutions that meet a minimum feasibility threshold.

Our evaluation was conducted using Gemini 2.5 Pro. To ensure deterministic and fully reproducible results, the temperature was set to 0, and a single definitive solution was generated for each baseline.

### 5.1.3 BASELINES

In our evaluation benchmark, we compare UoT against several established reasoning techniques:

- *Zero-Shot Prompting*: The task description was provided directly to the LLM.
- *Chain-of-Thought (CoT)* (Wei et al., 2022): The model was prompted to generate a step-by-step reasoning process.
- *Tree-of-Thoughts (ToT)* (Yao et al., 2023): A method where the model explores a tree of branching reasoning paths.
- *Enhanced Graph-of-Thoughts (EGoT)* (Shin & Kim, 2025): A graph-based approach that allows for merging and refining intermediate thoughts.

We used GPT-4o as the primary backbone model for implementing all baseline methods. Moreover, just to situate our results versus current closed proprietary models, we also compared the outputs generated by our framework on GPT-4o against state-of-the-art proprietary models, including *GPT-5* (OpenAI, 2025) and *DeepSeek V3.1* (Liu et al., 2024), with the understanding that they might already include (unknown) creative reasoning capabilities.

### 5.2 RESULTS

Tables 3 summarizes results of our experimentation. We generated 10 unique solutions for each method; the final reported score is the average of 10 runs. Our results demonstrate the effectiveness of the UoT methods. Across all tasks, the UoT variants—particularly T-UoT and C-UoT—consistently achieve the highest creativity scores among all reasoning methods. Notably, UoT closes the performance gap with more powerful proprietary models. For instance, with the Bridge Task, T-UoT (Creativity: 0.698) significantly outperforms GPT-5 (Creativity: 0.649), showcasing that a specialized reasoning structure can elicit superior creative performance even from a less advanced backbone model. While DeepSeek V3.1 shows the strongest performance on the

Table 3: Performance comparison across all creative reasoning tasks. C=Creativity, N=Novelty, U=Utility, and F=Feasibility.

| Method | Bridge Task | | | | Electricity Task | | | | Society Task | | | |
|---|---|---|---|---|---|---|---|---|---|---|---|---|
| | C | N | U | F | C | N | U | F | C | N | U | F |
| Zero Shot | 0.566 | 0.732 | 0.400 | 0.925 | 0.508 | 0.587 | 0.430 | 0.680 | 0.522 | 0.675 | 0.370 | 0.730 |
| COT | 0.651 | 0.742 | 0.560 | 0.980 | 0.450 | 0.541 | 0.360 | 0.675 | 0.586 | 0.732 | 0.440 | 0.560 |
| TOT | 0.579 | 0.799 | 0.360 | 0.900 | 0.488 | **0.657** | 0.320 | 0.610 | 0.574 | 0.727 | 0.420 | 0.495 |
| EGOT | 0.641 | 0.643 | 0.640 | **1.000** | 0.409 | 0.428 | 0.390 | **0.825** | 0.613 | 0.775 | 0.450 | 0.525 |
| C-UoT | 0.664 | 0.728 | 0.600 | **1.000** | 0.512 | 0.563 | 0.460 | 0.675 | 0.649 | 0.799 | 0.500 | 0.530 |
| E-UoT | 0.589 | 0.778 | 0.400 | 1.000 | 0.463 | 0.587 | 0.340 | 0.755 | 0.545 | 0.810 | 0.280 | 0.575 |
| T-UoT | **0.698** | **0.807** | 0.590 | 0.930 | 0.526 | 0.641 | 0.410 | 0.665 | 0.628 | **0.846** | 0.410 | 0.575 |
| GPT-5 | 0.649 | 0.649 | **0.650** | **1.000** | 0.504 | 0.497 | 0.510 | 0.815 | 0.631 | 0.652 | 0.610 | **0.870** |
| DeepSeek V3.1 | 0.458 | 0.476 | 0.440 | 0.950 | **0.571** | 0.563 | **0.580** | 0.770 | **0.723** | 0.767 | **0.680** | 0.825 |

Electricity and Society Tasks, the ability of UoT to compete with and surpass state-of-the-art models underscores its significance.

A closer look at UoT variants reveals performance trade-offs that align with their design principles. As expected, novelty scores generally increase from C-UoT to E-UoT to T-UoT, corresponding with the greater conceptual freedom at each level. For example, with the Society Task, T-UoT achieves the highest novelty score (0.846). However, this pursuit of novelty highlights a crucial challenge for E-UoT, which consistently struggles with utility, scoring just 0.280 on the Society Task despite its high novelty of 0.810.

Table 4 in Appendix C.1 presents representative solutions from each UoT method, revealing distinct creative processes. C-UoT demonstrates successful analogical transfer, identifying a relevant concept—"negotiation protocols in distributed systems"—from a parallel engineering problem domain and applying it to the traffic problem. This approach is effective because it leverages a proven solution structure from a structurally similar problem.

E-UoT generates a valid exploratory solution by incorporating "real-time traffic data" into a dynamic feedback loop. While this method is adaptive, its iterative, localized adjustments may struggle to achieve a global optimum, which is reflected in its lower utility score compared to more holistic approaches.

Most notably, T-UoT has produced a highly novel combinational solution. During its process, T-UoT generates genuinely transformative rule sets (e.g., reframing vehicle headway as an "energy barrier to overcome"). However, when tasked with producing a final, high-utility solution, it selected the quantum annealing approach. This suggests a potential novelty-utility tradeoff: the most abstract and transformative rule changes may be difficult to ground in a practical, high-scoring solution, leading the model to converge on a less transformative but more novel and effective combinational idea. This finding highlights the inherent difficulty in generating solutions that are both fundamentally rule-breaking and immediately optimal.

## 6 Conclusions and Future Work

In this work, we introduced a computational framework for creative reasoning grounded in cognitive science. We instantiated this framework with the Universe of Thoughts (UoT), a novel set of methods that enables Large Language Models to move beyond conventional problem-solving. To rigorously assess this capability, we also developed a new evaluation benchmark with open-ended tasks requiring both creative and logical reasoning.

Our empirical results show that UoT significantly outperforms existing reasoning methods. Remarkably, when implemented with GPT-4o, UoT generated solutions rated higher than those from a more advanced proprietary model, GPT-5. This finding underscores that a superior creative reasoning framework can elevate the performance of an existing model beyond that of a more powerful model operating without such guidance.

This research opens the door for AI systems that do not just solve predefined problems, but creatively discover novel solutions. In the future, we will further explore other methods to instantiate our propose computational framework for creative reasoning.

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

## A  APPENDIX: FORMALIZATION

### A.1  COMBINATIONAL UOT (C-UOT)

**Settings**    **Definition 1 (Problem Space).** Let $\mathcal{P}$ denote the space of all problems under consideration.

**Definition 2 (Rule Set).** For each problem $P \in \mathcal{P}$, let $\mathcal{R}_P = \{r_1, r_2, \ldots, r_k\}$ be the rule set associated with problem $P$, where each $r_i : \mathcal{S} \to \{0, 1\}$ is a constraint function that determines whether a solution satisfies the $i$-th rule.

**Definition 3 (Solution Library).** For a given problem $P \in \mathcal{P}$, let $\mathcal{K}_P = \{s_{P,1}, s_{P,2}, \ldots, s_{P,n}\}$ denote the library of known solutions to problem $P$.

**Definition 4 (Solution Decomposition).** Each solution $s_{P,i} \in \mathcal{K}_P$ can be decomposed into a finite set of constituent thoughts:

$$\mathcal{C}(s_{P,i}) = \{c_1, c_2, \ldots, c_{m_i}\}$$

where $c_j$ represents an atomic idea.

**Definition 5 (Embedding Functions).** We define two types of embedding functions that map solutions and thoughts to vector spaces:

- **Functional/Role Embedding**: $\phi_{\text{role}} : \mathcal{S} \cup \mathcal{C} \to \mathbb{R}^{d_1}$ captures the functional role or purpose of solutions and thoughts

- **Surface/Context Embedding**: $\psi_{\text{surf}} : \mathcal{S} \cup \mathcal{C} \to \mathbb{R}^{d_2}$ captures surface-level features and contextual information

where $\mathcal{S} = \bigcup_{P \in \mathcal{P}} \mathcal{K}_P$ is the set of all solutions and $\mathcal{C} = \bigcup_{s \in \mathcal{S}} \mathcal{C}(s)$ is the set of all constituent thoughts.

**Definition 6 (Feasibility and Utility Functions).** Let $F_{\mathcal{R}_P} : \mathcal{S} \to \{0, 1\}$ be a feasibility function that determines whether a solution is feasible under rule set $\mathcal{R}_P$, and $U : \mathcal{S} \to \mathbb{R}$ be a utility function that assigns a real-valued utility to each solution.

**Definition 7 (Similarity and Distance Metrics).** Let $\text{sim} : \mathbb{R}^d \times \mathbb{R}^d \to [0, 1]$ be a similarity function and $\text{dist} : \mathbb{R}^d \times \mathbb{R}^d \to \mathbb{R}_{\geq 0}$ be a distance function, where these metrics operate on the embedded representations produced by $\phi_{\text{role}}$ and $\psi_{\text{surf}}$.

**Step 1 (Analogical Retrieval & Solution Harvesting).**    Let $P_t \in \mathcal{P}$ be a target problem. Define the analogical problem set as:

$$\mathcal{P}_{\text{analog}} = \{P \in \mathcal{P} : \text{sim}(\phi_{\text{role}}(P), \phi_{\text{role}}(P_t)) \geq \tau_{\text{role}} \text{ and } \text{dist}(\psi_{\text{surf}}(P), \psi_{\text{surf}}(P_t)) \geq \tau_{\text{surf}}\}$$

where $\tau_{\text{role}} > 0$ is a similarity threshold for role embeddings and $\tau_{\text{surf}} > 0$ is a distance threshold for surface embeddings.

Collect all solutions from the analogical problems:

$$\mathcal{S}_{\text{harvest}} = \bigcup_{P \in \mathcal{P}_{\text{analog}}} \mathcal{K}_P$$

**Step 2 (Decompose Each Solution into Thoughts).** Decompose each solution $s \in \mathcal{S}_{\text{harvest}}$ using the decomposition function from Definition 4 to obtain the complete set of constituent thoughts:

$$\mathcal{C}_{\text{harvest}} = \bigcup_{s \in \mathcal{S}_{\text{harvest}}} \mathcal{C}(s)$$

where $\mathcal{C}(s)$ denotes the set of atomic thoughts comprising solution $s$.

**Step 3 (Choose a Host & Substitution Sites).** Select a diverse subset of host solutions from the target problem's solution library:

$$\mathcal{H} \subseteq \mathcal{K}_{P_t}$$

such that $\mathcal{H} = \{h_1, h_2, \ldots, h_m\}$ maximizes diversity in surface embedding space:

$$\mathcal{H} = \arg \max_{\mathcal{H}' \subseteq \mathcal{K}_{P_t}, |\mathcal{H}'|=m} \sum_{i<j} \text{dist}(\psi_{\text{surf}}(h_i'), \psi_{\text{surf}}(h_j'))$$

For each host solution $h_i \in \mathcal{H}$, decompose it into constituent thoughts $\mathcal{C}(h_i) = \{c_1^{(i)}, c_2^{(i)}, \ldots, c_{k_i}^{(i)}\}$ and select substitution sites based on idea impact:

$$\mathcal{I}_{\text{sub}}^{(i)} \subseteq \mathcal{C}(h_i)$$

where substitution sites are chosen to maximize creative potential while maintaining solution feasibility.

**Step 4 (Far-then-Analogical Donor Selection).** For each host solution $h_i \in \mathcal{H}$ and each idea $c_j^{(i)} \in \mathcal{I}_{\text{sub}}^{(i)}$ selected for substitution, identify the top-$k$ donor thoughts from $\mathcal{C}_{\text{harvest}}$:

$$\mathcal{D}_j^{(i)} = \arg \max_{D \subseteq \mathcal{C}_{\text{harvest}}, |D|=k} \sum_{d \in D} \left[ \text{sim}(\phi_{\text{role}}(d), \phi_{\text{role}}(c_j^{(i)})) - \lambda \cdot \text{sim}(\psi_{\text{surf}}(d), \psi_{\text{surf}}(c_j^{(i)})) \right]$$

where $\lambda > 0$ is a weighting parameter that balances role similarity maximization with surface similarity minimization, effectively selecting donor thoughts that are functionally analogous but superficially distant from the original idea $c_j^{(i)}$.

**Step 5 (Substitute to Synthesize New Combinations).** For each host solution $h_i \in \mathcal{H}$, generate new candidate solutions by systematically substituting thoughts from their corresponding donor sets. For each substitution site $c_j^{(i)} \in \mathcal{I}_{\text{sub}}^{(i)}$ and each donor idea $d \in \mathcal{D}_j^{(i)}$, create a new solution:

$$s_{i,j,d}' = (h_i \setminus \{c_j^{(i)}\}) \cup \{d\}$$

where the substitution replaces the original idea $c_j^{(i)}$ with donor idea $d$.

The complete set of synthesized candidate solutions is:

$$\mathcal{S}_{\text{new}} = \bigcup_i \bigcup_{j: c_j^{(i)} \in \mathcal{I}_{\text{sub}}^{(i)}} \bigcup_{d \in \mathcal{D}_j^{(i)}} \{s_{i,j,d}'\}$$

**Step 6 (Evaluate and Rank by Feasibility, Usefulness, Novelty).** For each candidate solution $s' \in \mathcal{S}_{\text{new}}$, compute a composite evaluation score:

$$E(s') = \alpha \cdot F_{\mathcal{R}_{P_t}}(s') + \beta \cdot U(s') + \gamma \cdot N(s')$$

where $F_{\mathcal{R}_{P_t}}(s') \in \{0, 1\}$ is the feasibility function from Definition 6, $U(s')$ is the utility function, $N(s')$ is a novelty measure, and $\alpha, \beta, \gamma \geq 0$ are weighting parameters with $\alpha + \beta + \gamma = 1$.

Define the novelty measure as:

$$N(s') = 1 - \max_{s \in \mathcal{K}_{P_t}} \text{sim}(\phi_{\text{surf}}(s'), \phi_{\text{surf}}(s))$$

Rank all candidate solutions by their evaluation scores and select the final solution set:

$$\mathcal{S}_{\text{final}} = \{s' \in \mathcal{S}_{\text{new}} : E(s') \geq \tau_{\text{eval}}\}$$

where $\tau_{\text{eval}} > 0$ is an evaluation threshold for solution acceptance.

## A.2 EXPLORATORY UoT (E-UoT)

**New Step E (Exploratory Idea Expansion; insert after Step 2).** Select a diverse subset of seed thoughts from the harvested thoughts to maximize surface-level diversity:

$$\mathcal{C}_{\text{seed}} = \arg \max_{\mathcal{C}' \subseteq \mathcal{C}_{\text{harvest}}, |\mathcal{C}'| = \ell} \sum_{i < j} \text{dist}(\psi_{\text{surf}}(c_i'), \psi_{\text{surf}}(c_j'))$$

where $\ell$ is the number of seed thoughts to select.

For each seed idea $c_{\text{seed}} \in \mathcal{C}_{\text{seed}}$, discover new analogical thoughts by finding thoughts that are functionally similar but superficially distant:

$$c_{\text{new}} = \arg \max_{c \in \mathcal{C} \setminus \mathcal{C}_{\text{harvest}}} [\text{sim}(\phi_{\text{role}}(c), \phi_{\text{role}}(c_{\text{seed}})) - \lambda \cdot \text{sim}(\psi_{\text{surf}}(c), \psi_{\text{surf}}(c_{\text{seed}}))]$$

where $\mathcal{C}$ is the universal set of all possible thoughts and $\lambda > 0$ is a weighting parameter.

Expand the harvested idea set with the newly discovered thoughts:

$$\mathcal{C}_{\text{harvest}} \leftarrow \mathcal{C}_{\text{harvest}} \cup \bigcup_{c_{\text{seed}} \in \mathcal{C}_{\text{seed}}} \{c_{\text{new}}(c_{\text{seed}})\}$$

## A.3 TRANSFORMATIVE UoT (T-UoT)

**Step 1 (Expose Rules, Including Hidden Assumptions).** For the target problem $P_t \in \mathcal{P}$, construct the complete rule set by identifying both explicit and implicit constraints:

$$\mathcal{R}_{P_t}^{\text{complete}} = \mathcal{R}_{P_t}^{\text{explicit}} \cup \mathcal{R}_{P_t}^{\text{hidden}}$$

where $\mathcal{R}_{P_t}^{\text{explicit}}$ represents the explicitly stated rules and $\mathcal{R}_{P_t}^{\text{hidden}}$ represents hidden assumptions that implicitly constrain the solution space.

The hidden assumptions can be discovered through analysis of existing solutions:

$$\mathcal{R}_{P_t}^{\text{hidden}} = \{r : \mathcal{S} \to \{0, 1\} \mid \forall s \in \mathcal{K}_{P_t}, r(s) = 1 \text{ and } r \notin \mathcal{R}_{P_t}^{\text{explicit}}\}$$

This process exposes implicit constraints that may unnecessarily limit the solution space, enabling more creative exploration by potentially relaxing or challenging these hidden assumptions.

**Step 2 (Rule-Mutation Process to Generate New Rule Sets).** **Analogical Rule Discovery:** For each rule $r \in \mathcal{R}_{P_t}^{\text{complete}}$, find analogical rules from other problem domains:

$$r_{\text{analog}} = \arg \max_{r' \in \bigcup_{P \neq P_t} \mathcal{R}_P} [\text{sim}(\phi_{\text{role}}(r'), \phi_{\text{role}}(r)) - \lambda \cdot \text{sim}(\psi_{\text{surf}}(r'), \psi_{\text{surf}}(r))]$$

**Impact-Based Rule Selection:** Select high-impact rules from the complete rule set:

$$\mathcal{R}_{\text{impact}} = \{r \in \mathcal{R}_{P_t}^{\text{complete}} : I(r) \geq \tau_{\text{impact}}\}$$

where $I : \mathcal{R}_{P_t}^{\text{complete}} \to \mathbb{R}_{\geq 0}$ measures the impact of rule $r$ on the target problem's solution space.

**Rule Mutation Operations:** For each selected rule $r \in \mathcal{R}_{\text{impact}}$, apply mutation operations:

- **Drop Operation:** $\mathcal{R}_{\text{drop}}^{(r)} = \mathcal{R}_{P_t}^{\text{complete}} \setminus \{r\}$
- **Vary Operation:** $\mathcal{R}_{\text{vary}}^{(r)} = (\mathcal{R}_{P_t}^{\text{complete}} \setminus \{r\}) \cup \{r_{\text{analog}}\}$

**Rule Addition:** For each mutated rule set $\mathcal{R}_{\text{mut}}$, generate expanded rule sets:

$$\mathcal{R}_{\text{add}}^{(\mathcal{R}_{\text{mut}})} = \mathcal{R}_{\text{mut}} \cup \{r_{\text{new}}\}$$

where $r_{\text{new}}$ is constrained to ensure feasibility: $\exists s \in \mathcal{S} : \forall r \in \mathcal{R}_{\text{add}}^{(\mathcal{R}_{\text{mut}})}, r(s) = 1$.

**Complete Collection:** The final collection of new rule sets is:

$$\mathcal{R}_{\text{collection}} = \bigcup_{r \in \mathcal{R}_{\text{impact}}} \{\mathcal{R}_{\text{drop}}^{(r)}, \mathcal{R}_{\text{vary}}^{(r)}\} \cup \bigcup_{\mathcal{R}_{\text{mut}}} \{\mathcal{R}_{\text{add}}^{(\mathcal{R}_{\text{mut}})}\}$$

**Step 3 (Explore Each New Rule Space for Solutions).** For each new rule set $\mathcal{R}_{\text{new}} \in \mathcal{R}_{\text{collection}}$, apply the Exploratory Creative Reasoning process by defining a modified target problem:

$$P_{\text{new}} = (P_t, \mathcal{R}_{\text{new}})$$

Execute the complete creative reasoning pipeline for each modified problem:

$$\text{ECR}(P_{\text{new}}) : P_{\text{new}} \mapsto \mathcal{S}_{\text{creative}}^{(P_{\text{new}})}$$

where $\text{ECR}(\cdot)$ denotes the Exploratory Creative Reasoning algorithm (Steps 1-6 from the main process) applied to problem $P_{\text{new}}$ with rule set $\mathcal{R}_{\text{new}}$.

Collect all solutions generated across the modified rule spaces:

$$\mathcal{S}_{\text{rule-exploration}} = \bigcup_{\mathcal{R}_{\text{new}} \in \mathcal{R}_{\text{collection}}} \text{ECR}(P_{\text{new}})$$

Filter solutions for feasibility under the original constraints while preserving rule-space diversity:

$$\mathcal{S}_{\text{final}} = \left\{ s \in \mathcal{S}_{\text{rule-exploration}} : F_{\mathcal{R}_{P_t}^{\text{complete}}}(s) = 1 \vee \text{CreativeValue}(s) \geq \tau_{\text{creative}} \right\}$$

where $\text{CreativeValue}(s)$ measures the creative potential of solutions that may violate original constraints.

# B   APPENDIX: COMPLEXITY ANALYSIS

We analyze the computational complexity of our UoT framework against a baseline of exhaustive search. Let's define the key parameters of the problem space:

- $d$: The number of problem domains.
- $s$: The number of known solutions per domain.
- $c$: The number of thoughts composing a single solution.
- $a$: The number of analogous domains considered by UoT ($a \ll d$).
- $o$: The number of outside thoughts not part of any known solution.
- $r$: The number of rules defining the solution space.

Let's also define efficiency parameters for UoT's guided search, where each $\eta < 1$ represents a significant reduction in search effort compared to brute force:

- $\eta_{analog}$: Efficiency of finding relevant analogous problems.
- $\eta_{synth}$: Efficiency of synthesizing thoughts into a coherent solution.
- $\eta_{rules}$: Efficiency of mutating and evaluating new rule sets.

## B.1   COMBINATIONAL UoT (C-UoT)

**Baseline (Exhaustive Search):** The total pool of thoughts available from all known solutions is $d \times s \times c$. An exhaustive search must try every possible combination of $c$ thoughts from this pool. The complexity is the number of permutations of choosing $c$ thoughts from the total pool:

$$O_{base\_C} = P(d \cdot s \cdot c, c) = \frac{(d \cdot s \cdot c)!}{(d \cdot s \cdot c - c)!}$$

**C-UoT:** C-UoT narrows the search to a few analogous domains ($a$) and then intelligently synthesizes a solution. Its complexity is a product of its guided search steps:

$$O_{C-UoT} = (a \cdot s \cdot c \cdot \eta_{analog}) \cdot \eta_{synth}$$

**Efficiency Gain:** The gain is the ratio of the baseline complexity to C-UoT's complexity.

$$\text{Gain}_C = \frac{O_{base\_C}}{O_{C-UoT}} = \frac{(d \cdot s \cdot c)!}{((d \cdot s \cdot c - c)!) \cdot (a \cdot s \cdot c \cdot \eta_{analog} \cdot \eta_{synth})}$$

Given that $d \gg a$ and the factorial growth of the baseline, the efficiency gain is astronomically large.

## B.2 EXPLORATORY UoT (E-UoT)

**Baseline (Exhaustive Search):** The baseline must now consider all known thoughts plus all outside thoughts, creating a total pool of $d \cdot s \cdot c + o$. The complexity is:

$$O_{base\_E} = P(d \cdot s \cdot c + o, c) = \frac{(d \cdot s \cdot c + o)!}{(d \cdot s \cdot c + o - c)!}$$

**E-UoT:** E-UoT's process is similar to C-UoT but includes an efficient search for novel outside thoughts. We can model this with the same parameters, as the search for outside thoughts is part of the guided synthesis process.

$$O_{E-UoT} = (a \cdot s \cdot c \cdot \eta_{analog}) \cdot \eta_{synth}$$

**Efficiency Gain:**

$$\text{Gain}_E = \frac{O_{base\_E}}{O_{E-UoT}} = \frac{(d \cdot s \cdot c + o)!}{((d \cdot s \cdot c + o - c)!) \cdot (a \cdot s \cdot c \cdot \eta_{analog} \cdot \eta_{synth})}$$

## B.3 TRANSFORMATIVE UoT (T-UoT)

**Baseline (Exhaustive Search):** The baseline must perform the full exploratory search for every possible combination and permutation of the $r$ rules. This adds a factorial term for exploring all rule mutations, leading to a massive complexity:

$$O_{base\_T} = P(d \cdot s \cdot c + o, c) \times r! = \frac{(d \cdot s \cdot c + o)!}{(d \cdot s \cdot c + o - c)!} \cdot r!$$

**T-UoT:** T-UoT adds an efficient step for mutating and selecting a new rule set to the exploratory process:

$$O_{T-UoT} = (a \cdot s \cdot c \cdot \eta_{analog}) \cdot \eta_{synth} \cdot \eta_{rules}$$

**Efficiency Gain:**

$$\text{Gain}_T = \frac{O_{base\_T}}{O_{T-UoT}} = \frac{(d \cdot s \cdot c + o)! \cdot r!}{((d \cdot s \cdot c + o - c)!) \cdot (a \cdot s \cdot c \cdot \eta_{analog} \cdot \eta_{synth} \cdot \eta_{rules})}$$

The introduction of the $r!$ term makes the efficiency gain of T-UoT over a true exhaustive baseline the most significant of the three methods.

# C SUPPLEMENTARY RESULTS

## C.1 SAMPLE SOLUTIONS FOR BRIDGE TASK

## C.2 SENSITIVITY ANALYSIS

Figures 2a and 2b present sensitivity of C-UoT's performance on the Bridge Task with respect to two key parameters: the number of analogous problems and the number of solutions generated per problem.

Figure 2a shows that performance follows a U-shaped curve as the number of analogous problems increases. We hypothesize this indicates three distinct regimes: a small number of analogies may yield a few high-quality, easily combinable thoughts; a moderate number can introduce conceptual noise that hinders effective synthesis; and a very large number provides a sufficiently rich and diverse pool of concepts where the benefits of flexibility overcome the noise.

Figure 2b shows that increasing the number of solutions per analogous problem, while exhibiting some fluctuation, generally leads to improved performance. This suggests that a larger pool of "conceptual primitives" or thoughts derived from each analogy provides more raw material for synthesizing a high-quality final solution.

Table 4: Sample Solutions for Bridge Task

| Method | Solution |
|--------|----------|
| C-UoT | The core functional mechanism is a cooperative negotiation system. Its essential components are a driver interface for signaling crossing intent (input) and a central allocation system. The system aggregates these individual intents to form a comprehensive demand profile, and based on this, allocates specific crossing times (output) back to the drivers. This interaction, inspired by negotiation protocols in distributed systems, ensures fair and efficient resource use. |
| E-UoT | Based on a feedback-based mechanism, this solution implements a dynamic scheduling model that uses real-time traffic data (input) to continuously adjust vehicle crossing schedules. This interaction dynamically allocates crossing windows based on current conditions (output), which serves to optimize overall flow and reduce vehicle idle times. |
| T-UoT | The functional core is an optimization algorithm that applies the principles of quantum annealing to traffic scheduling. The problem of finding the best crossing schedule is formulated as an energy minimization problem. The algorithm then simultaneously explores a vast landscape of possible schedule configurations to identify the global optimum schedule as output, which minimizes overall delay. This approach, inspired by quantum computing, is designed to efficiently solve complex combinatorial optimization problems. |

(a) Effect of number of analogous problems    (b) Effect of number of solutions generated

Figure 2: *Sensitivity analysis for UoT*

## D    THE USE OF LARGE LANGUAGE MODELS (LLMS)

We leverage Large Language Models (LLMs) for grammar correction and typo elimination to polish our writing.

