# OpenReview forum: "Universe of Thoughts: Enabling Creative Reasoning with Large Language Models"
_ICLR.cc/2026/Conference — Submitted to ICLR 2026_

### Official Review · Reviewer_vU6P · 2025-10-28

**Soundness:** 2
**Presentation:** 2
**Contribution:** 2
**Rating:** 2
**Confidence:** 3

**Summary:**

This paper addresses the lack of autonomous creative reasoning in existing LLM reasoning frameworks. It draws on Boden's three creativity types from cognitive science to propose a computational framework for creative reasoning, instantiated as UoT (including C-UoT, E-UoT, and T-UoT). An evaluation benchmark with 3 open-domain tasks is constructed, using feasibility, utility, and novelty as metrics to compare against traditional frameworks and commercial models. Experiments show UoT (especially T-UoT) achieves the highest creativity scores, even enabling GPT-4o to outperform GPT-5, offering a new direction for autonomous creative reasoning in LLMs.

**Strengths:**

- It is the first to translate Boden's creativity theory into a deployable computational framework for LLMs.
- UoT's three variants have a logical progression and modular reproducibility.
- The evaluation benchmark covers multiple domains with scientific metrics; it also introduces solution canonicalization to eliminate evaluation bias, ensuring reusability.
- Comprehensive experiments clearly demonstrate UoT's advantages and validate that "framework design outperforms model scale."

**Weaknesses:**

- Validation is limited to custom tasks (no public creative reasoning datasets are used). Analysis of key parameter impact boundaries is insufficient, leaving generalization and robustness unproven.
- Most high-scoring solutions of T-UoT are combinatorial innovations, failing to achieve its core goal of "rule-breaking." The effectiveness threshold for rule mutation is not analyzed.
- Utility and novelty evaluations rely on LLM subjective judgments (no human annotation verification). The criteria for constructing the "known solution set" are unclear, raising doubts about objectivity.
- Computational complexity is only derived theoretically; no experimental comparisons of reasoning time or resource consumption are provided, leaving efficiency advantages in large-scale scenarios unsubstantiated.

**Questions:**

- What are the criteria for selecting analogous problems in UoT's "idea pool construction"? How to avoid analogy bias and ensure automated selection?
- How does E-UoT verify the validity of new ideas? Is there a mechanism to filter irrelevant or invalid ideas?
- In the social cohesion task, how are metrics like "new ties" and "mixing" quantified? Are they based on simulation or real data?
- When comparing with GPT-5, why are its reasoning settings (e.g., use of prompt engineering) unstated? How to rule out the possibility of GPT-5 integrating UoT-like mechanisms internally?
- In complex fields (e.g., drug discovery), does UoT require domain knowledge injection? How to balance domain constraints and creativity?
- Are there solutions to E-UoT's "high novelty but low utility" issue, such as introducing feedback mechanisms or reinforcement learning?
- The necessity of the progressive order of UoT's three paradigms is unvalidated. Are there scenarios where skipping C-UoT still yields high performance? Can paradigm priorities be adjusted dynamically?
- Since T-UoT's utility does not outperform C-UoT, is rule-breaking not an optimal strategy for current LLMs, or do evaluation metrics fail to capture its long-term value?

---

> ### Author Response · Authors · 2025-11-25
>
> Thank you for your detailed comments and questions. Please find our answers to the posed questions below. Regarding the raised weaknesses, due to space constraints, we refer the reviewer to our response to Comment W3 from Reviewer vz22 regarding human evaluation, and to Comments W1 and Q1 of the Reviewer FMjA regarding generalizability of our approach.
>
> Q1) Analogous Problem Selection: We employ a dual-embedding strategy (detailed in the Appendix) to automate the selection of "Far Transfer" analogies. To avoid superficial bias, we project problems into two spaces: Surface-Level (keywords) and Role-Level (abstract structure). Our algorithm explicitly selects candidates with high Role-Level similarity but low Surface-Level similarity. This mathematically decouples abstract relationships from linguistic overlap, ensuring the retrieval of structurally relevant but domain-divergent inspirations.
>
> Q2) E-UoT Idea Verification: Validity is enforced implicitly during the retrieval phase via our dual-embedding mechanism (detailed in the Appendix):
>
> - Anchoring to Validity: We treat the manually verified "seed" thoughts as the ground truth for functional logic.
> - Role-Preservation as Filtering: When generating new thoughts, we filter for thoughts that possess high Role-Level similarity to these valid seeds. This metric acts as a strict gatekeeper: it rejects irrelevant or logically unsound ideas that do not align with the functional structure of the solution, ensuring that the retrieved thoughts differ only in surface domain (novelty) while retaining functional logic (validity).
>
> Q3) Social Task Evaluation: We employed an LLM-based semantic simulation to estimate these metrics. This was a deliberate design choice over rigid algorithmic simulation for two reasons:
>
> - Handling Out-of-Distribution Solutions: Algorithmic simulators require pre-defined variables and interaction rules. However, creative solutions often introduce novel mechanisms that strict mathematical models cannot capture.
> - Semantic vs. Numeric Evaluation: With open-ended social tasks, cohesion is often a qualitative outcome derived from complex interactions. An LLM evaluator can assess the causal chain of a proposed policy to estimate ``mixing'' and ``new ties'' based on reasoning, offering a more flexible and holistic assessment than a fixed heuristic could provide.
>
> Q4) Comparison with GPT-5: We employed standard zero-shot prompting for GPT-5 to evaluate its intrinsic capabilities. While proprietary models may utilize latent reasoning, they remain opaque ``black boxes,'' preventing them from contributing to the state-of-the-art knowledge. Our results demonstrate that UoT’s explicit, interpretable mechanism (specifically, Role-Level analogical transfer) provides a targeted cognitive advantage that outperforms generic model scaling on constrained creative tasks.
>
> Q5) Complex Domain Application: Yes, for specialized domains like drug discovery, domain knowledge is injected via the initial prompts and seed solutions. However, the core mechanism remains unchanged: our framework treats domain constraints as the ``Utility'' component of the creativity equation (Creativity = Novelty + Utility). The balance is enforced during the selection phase, where we rigorously rank solutions to ensure they strictly satisfy domain-specific utility constraints before optimizing for novelty.
>
> Q6) Feedback Mechanism: We agree that balancing utility is critical. Our framework creates a modular structure where this can be addressed by integrating an iterative self-correction loop during the Combinatorial Search phase. By introducing an intermediate ``Utility Check'' step (potentially via self-reflection or a learned reward model) before the final ranking, we can filter out high-novelty/low-utility thoughts early in the generation process, ensuring that the final output remains functional.
>
> Q7) Framework Relationship: We clarify that the three paradigms are not a rigid progressive sequence. They are independent in a modular framework. However, they share a hierarchical relationship: C-UoT serves as a fundamental operator that is utilized as a sub-module within the broader E-UoT and T-UoT architectures to facilitate concept blending.
>
> Q8) T-UoT Utility: We attribute this observation to the inherent tension between novelty and utility in current LLMs, rather than a failure of the strategy:
> - Generator Limitation (Out-of-Distribution): T-UoT forces the model to break rules and operate in low-probability regions of its training distribution. Current models struggle to ground these radically ``out-of-distribution'' ideas in practical logic (utility) compared to safer, conventional solutions.
>  - Evaluator Bias: We hypothesize that LLM-based evaluators exhibit a conservative bias. They tend to rate familiar solutions (high precedence) as more ``useful'' and may unfairly penalize valid but radically novel solutions simply because they deviate from established patterns.

---

### Official Review · Reviewer_vz22 · 2025-10-29

**Soundness:** 2
**Presentation:** 2
**Contribution:** 2
**Rating:** 2
**Confidence:** 4

**Summary:**

This paper proposes Universe of Thoughts (UoT), a conceptual framework that enables creative reasoning in Large Language Models (LLMs). Drawing from Margaret Boden’s taxonomy of creativity, the authors define three reasoning paradigms, such as combinational, exploratory, and transformative reasoning, and outline how these processes can be instantiated in LLMs. A new benchmark with open-ended tasks is introduced to evaluate creativity using metrics of novelty, utility, and feasibility. Experiments show UoT outperforming Chain-of-Thought (CoT), Tree-of-Thought (ToT), and commercial reasoning models on these self-designed tasks.

**Strengths:**

Strengths:

- This paper tackles an interesting topic, which formalizes “creative reasoning” for LLMs.

- By grounding the conceptual framework in cognitive science (Boden’s theory), this paper establishes a solid theoretical background for creative reasoning.

- The proposed benchmark may inspire further exploration of creativity evaluation in reasoning systems.

**Weaknesses:**

Weaknesses:

- Limited Novelty and Positioning. The notion of enabling creativity in LLMs is not new. Substantial prior work has investigated open-ended or ill-posed tasks, such as creative writing [5], story generation [7], and idea synthesis [6]. The paper does not adequately engage with this literature or clarify how its proposal differs conceptually or methodologically.

- Missing Discussion of Established Reasoning Paradigms. The manuscript omits connections to analogical reasoning [1,2], inductive reasoning [4], and deductive reasoning [3], which are well-studied cognitive and computational paradigms that are central to creativity and generalization. Without situating UoT within or against these established reasoning forms, its theoretical framing of “creative reasoning” feels incomplete and insufficiently contextualized.

- Evaluation Limitations. The benchmark is self-constructed and narrow in scope, which raises reproducibility and bias concerns.

[1]. Thought propagation: an analogical approach to complex reasoning with large language models. ICLR 2024.

[2]. Large language models are analogical reasoners. ICLR 2024.

[3]. Inductive or deductive rethinking the fundamental reasoning abilities of llms.

[4]. Hypothesis search inductive reasoning with language models. ICLR 2024.

[5]. Deliberate Problem Solving with Large Language Models. NeurIPS 2024.

[6]. Moose chem:  large language models for rediscovering unseen chemistry scientific hypotheses. ICLR 2025.

[7]. Overview of Long Story Generation Challenge (LSGC) at INLG 2024. ACL 2024.

**Questions:**

The authors are encouraged to address the concerns in Weaknesses.

---

> ### Author Response · Authors · 2025-11-25
>
> Thank you for your detailed feedback. Please find our response below for your kind review.
>
> W1 and W2) Lack of Novelty: (Please also review our response to the relevant Comment W3 from Reviewer cj1x) We acknowledge that our original submission could have covered the literature on reasoning more comprehensively, whereas we mostly focused on the related work we found closest to our work. With the revision, we will significantly expand the related work section to include the referenced line of work. However, here we emphasize that to the best of our knowledge, UoT is the ``first framework to introduce an explicit computational mechanism designed to systematically engineer creative solutions'', rather than relying on stochastic solution generation and/or human-crafted prompts. Below we elaborate on the unique properties of UoT, which we believe make it the first framework to address creative reasoning systematically:
>
> - Constrained Problem Solving vs. Unconstrained Generation( as in references [4], [6], and [7] in the reviewer's comments): While prior work explores creativity in unconstrained settings (e.g., story generation, creative writing), our work targets constrained open-ended problem solving. In particular, in tasks like ``Idea Synthesis,'' the goal is often merely novelty or fluency. In our setting, the generated solution must satisfy a dual objective: it must be not only novel but also functional (i.e., it must solve a specific problem). This adds a layer of complexity—to ensure utility—that unconstrained generation methods do not address.
> - Creative Engineering vs. Stochastic Prompting (as in references [1], [2], and [5] in the reviewer's comments): Existing methods (e.g., the standard Tree of Thoughts method applied to writing) rely heavily on the inherent stochasticity of the LLM or human-crafted prompts to stumble upon creative outputs. In contrast, UoT proposes and follows a deterministic mechanism for creativity. By algorithmically enforcing the retrieval of analogous problems, decomposing them into ``thoughts,'' and recombining them, we engineer the creative process rather than waiting for it to emerge by chance or rely on human creative thinking obtained through prompts.
> - Distinction from Standard Reasoning Paradigms (as in references [1-3] in the reviewer's comments): We agree that analogical and deductive reasoning are central to AI. However, standard implementations of these paradigms are typically convergent, i.e., they aim to find the single most logical answer. Our framework utilizes these reasoning forms as sub-modules but orchestrates them in a divergent-convergent workflow (inspired by Boden’s taxonomy) to specifically optimize for novelty, a metric that standard logical reasoning frameworks do not prioritize.
>
>
> W3) Evaluation: To the best of our knowledge, since we are the first to introduce systematic creative reasoning, we had to also introduce the first benchmark for systematic creativity. Unlike existing datasets that focus on constrained tasks or require heavy domain knowledge, our benchmark provides open-ended problems with an expansive solution space. We hope this serves as the preliminary step toward developing a full-scale benchmark in the future. To ensure reproducibility, all evaluations used temperature=0 with code and prompts fully disclosed. We also emphasize that our framework is task-agnostic, as further discussed in our reply to Commment W1 from Reviewer FMjA.
>
> To test the robustness of the evaluation, we conducted a human evaluation (N=10) using identical prompts to the LLM judge. Results align with LLM evaluation: Transformational UoT excels in physical tasks, while commercial models perform better in vague societal scenarios. While human-LLM correlation for Creativity is positive, it is lower for Feasibility; we attribute this to the high cognitive load required for humans to mentally simulate complex constraints compared to the LLM judge.
>
>
> * **C**: Creativity, **N**: Novelty, **U**: Utility, **F**: Feasibility
>
> |Method|Bridge C|Bridge N|Bridge U|Bridge F|Elec C|Elec N| Elec U|Elec F|Soc C|Soc N|Soc U|Soc F|
> | :--- | :---: | :---: | :---: | :---: | :---: | :---: | :---: | :---: | :---: | :---: | :---: | :---: |
> |**Zero Shot**|0.48|0.45|0.50|0.82|0.54|0.48|*0.61*|0.87|0.55|0.49| 0.61|**0.81**|
> |**CoT**|0.59|0.61|0.57|0.75|0.56|0.52|0.60|**0.92**|0.50|0.43|0.58|**0.81**|
> |**ToT**|0.58|0.60|0.56|0.68|0.53|0.54|0.52|0.76|0.53|0.44|0.61|**0.81**|
> |**EGOT**|*0.65*|*0.65*|*0.65*|0.81|0.50|0.44|0.55|*0.880*|0.56|0.48|*0.64*|0.80|
> |**C-UoT**|0.58|0.55|0.61|0.82|0.44|0.44|0.44|0.87|0.560|0.50| 0.62|**0.81**|
> |**E-UoT**|0.60|0.60|0.61|0.79|0.43|0.38|0.47|0.74|0.550|0.48| 0.62|0.73|
> |**T-UoT**|**0.68**|**0.69**|**0.67**|0.77|**0.62**|**0.61**| **0.64**|0.80|0.55|0.49|0.60|0.73|
> |**GPT-5**|0.62|0.61|0.64|*0.84*|*0.58*|*0.60*|0.56|0.77|*0.64*| *0.68*|0.60|0.77|
> |**DeepSeek R1**|0.46|0.46|0.47|**0.91**|*0.58*|0.58|0.58| 0.75|**0.72**|**0.69**|**0.74**|**0.81**|

---

### Official Review · Reviewer_FMjA · 2025-10-31

**Soundness:** 3
**Presentation:** 3
**Contribution:** 3
**Rating:** 6
**Confidence:** 3

**Summary:**

This paper proposes a computational framework and algorithmic pipeline to enable creative reasoning in LLMs, moving beyond conventional approaches like CoT/ToT to systematically explore a broader “universe of thoughts”. Inspired by cognitive science, the authors seek to invoke combinational, exploratory, and transformational creativity and operationalize each via structured prompting workflows that first retrieve analogous domains, decompose solutions into building-block ‘thoughts’, mutate conceptual rules, and synthesize novel solutions. They further design a benchmark with three open-ended tasks (traffic control, energy demand shaping, and social cohesion) and evaluate novelty, utility, and feasibility, showing that their Universe-of-Thoughts framework outperforms CoT/ToT/GoT and even exceeds GPT-5 on certain creative tasks using a weaker GPT-4o model. The core insight is that structuring the search over solution spaces and rules is key for creative problem-solving.

**Strengths:**

Timely contribution to the recent area of improving creativity in LLMs, with new creativity benchmark task.

The proposed approach is sound and based on well-studied paradigm of creative reasoning.

Experiments show that with proposed approaches, weaker (GPT-4o) LLM can outperform stronger comparison (GPT-5) in the measure of creativity proposed by the authors

**Weaknesses:**

Generalization to other creative tasks remains to be seen. The proposed creativity improvement approach seems to be tailored for the benchmark tasks proposed.

The approach seems to rely mostly on special prompts to help evoke creativity in LLMs and hence has limited technical novelty beyond prompt engineering.

**Questions:**

Regarding generalizability, how would this approach be adapted to other well-studied generative tasks such as code generation? It would be helpful to show its utility performance in a practical application.

---

> ### Author Response · Authors · 2025-11-25
>
> Thank you for your insights. we would like to address the raised weaknesses and questions as follows.
>
> W1) Generalization: We respectfully argue that our UoT framework is fundamentally task-agnostic. Unlike reasoning frameworks like Tree of Thoughts (ToT) that often require task-specific prompt tuning, UoT utilizes the exact same prompt templates across all three distinct domains (Logistics, Economics, and Social Dynamics). We hope this serves as evidence of systematic coss-domain generalizability for our proposed solution. The generalizability is due to the underlying UoT mechanism—retrieving analogous problems, decomposing thoughts, and recombining them—that is a general cognitive process of creativity rather than a task-specific heuristic. Thus, we claim that by design this method generalizes to any open-ended problem without modification.
>
> W2) Technical Novelty: We emphasize that our contribution is an algorithmic framework, not a prompt engineering technique. While we utilize LLMs as computational modules, the novelty lies in the structured control flow that orchestrates the creative process, specifically:
>
> - Analogous Retrieval: Systematically sourcing cross-domain inspirations.
> - Granular Decomposition: Breaking solutions down into atomic ``thoughts.''
> - Combinatorial Search: Algorithmically recombining these thoughts to maximize novelty while maintaining utility.
>
>
> This multi-stage pipeline implements a formal cognitive mechanism (inspired by Boden’s Combinational Creativity) that operates above the level of individual prompts. The LLM serves merely as the execution engine for this higher-level logic. We will further highlight this distinction in the revision.
>
> Q1) Generalizability: We did not include standard code benchmarks in this specific study because they primarily measure implementation correctness and syntax adherence rather than creative problem solving, which is the focus of our proposed work, UoT. However, we confirm that UoT is task-agnostic and can be applied toward code generation (among other tasks/domains), but as a tool to design novel software systems and algorithms rather than generating syntactically correct code. While the short revision period may not allow for developing and demonstrating applicability of such a design tool based on UoT, here we elaborate how UoT can be used for code generation in this context.
>
> For complex software engineering tasks (such as system and algorithm design) requiring introduction of novel architectural patterns, UoT is directly applicable (without modification to the core mechanism) to generate high-level abstract solutions (the blueprint or pseudocode). Accordingly, in an automated code generator UoT would serve as the architect agent, creating the novel algorithmic approach, while a standard Code-LLM can be used as the implementer agent, translating those creative thoughts into executable syntax.

---

### Official Review · Reviewer_cj1x · 2025-11-02

**Soundness:** 2
**Presentation:** 3
**Contribution:** 3
**Rating:** 6
**Confidence:** 4

**Summary:**

The authors are inspired by cognitive theories and propose three creative reasoning methods:
1. combinational reasoning, that transfers and combines thoughts from analogous domains
2. exploratory reasoning, that  introduces novel thoughts to expand the existing problem space
3. transformative reasoning, that alters fundamental rules to create a new, transformed solution space

The three methods were implemented with a set of Universe of Thoughts (UoT) and show that UoT demonstrates superior performance than CoT, ToT, etc.

**Strengths:**

1. The paper studies an interesting problem about creative reasoning and proposed UoT with combinatory, exploratory, and transformative reasoning, which is intuitive and showed good results.
2. The UoT part is backed with detailed formalization and complexity analysis in the appendix.

**Weaknesses:**

The quality of both the constructed tasks and evaluation methods remain unclear.

1. I am not sure how much I can trust the experimental results as the quality of the set of thoughts, search space, environment, etc., for the three tasks (One-Lane Bridge, Electricity Tariff, Social Cohesion) needs further validation. I could imagine this requires significant amount of engineering/simulation work and  domain expert knowledge. However, no discussion about the the process and validation to ensure the quality in these three domains are provided.

2. The utility and feasibility of a solution rely on LLM as a judge, yet the paper did not provide justification to support the use with a small set of human validation. This is my major concern and I could consider lower or increase my score depending on the author's response to this question.

3. Another smaller problem is that the paper seem to overstate its stance in **bridging the LLM reasoning and cognitive creativity**. This paper seems to have missed a line of work from the LLM/NLP community that also tackles the creative problem solving ability of LLMs, such as alternative uses, unconventional thinking, etc. I encourage the authors to consider adding comparison of model performance or discussion about these works.

Just naming a few as a starting point:

[1] Divergent association task: Divergent creativity in humans and large language models, by Bellemare-Pepin et al.,

[2] Unconventional (physical) problem-solving: MacGyver: Are Large Language Models Creative Problem Solvers? by Tian et al.,

[3] Idea generation: Can LLMS generate novel research ideas? A large-scale human study with 100+ nlp researchers, by Si et al.,

[4] There is a nice summary in: Large language models show both individual and collective creativity comparable to humans, by Sun et al.

**Questions:**

How are the thoughts, solutions, and existing solution space constructed for the three tasks? Where does the seeds come from and how do you ensure the accuracy and comprehensiveness?

---

> ### Author Response · Authors · 2025-11-25
>
> Thank you for your detailed comments. Please find our response below.
>
> W1) Experimental Design: We acknowledge the reviewer's concern regarding lack of sufficient information on the validation process used for the proposed evaluation tasks. We wish to clarify that while these tasks do not rely on external domain simulators (e.g., physics engines), they underwent a rigorous human-in-the-loop verification process to ensure logical consistency and robustness as follows:
>
> Regarding Task Construction and Validation Process: To ensure high quality without introducing confounding domain knowledge, we employed a multi-stage design process:
>
> - Candidate Generation: We initially generated ten candidate open-ended tasks.
> - Manual Solver Verification: The authors manually solved each candidate task to verify that the problem statements were logically sound and that the solution space was accessible.
> - Selection & Refinement: We selected the final three tasks (One-Lane Bridge, Electricity Tariff, Social Cohesion) based on their diversity. We then manually refined the rules to optimize the trade-off between strict constraints (to allow for objective evaluation) and open-endedness (to allow for creative reasoning).
>
>
> Regarding Rationale for Logic-Driven Constraints: As mentioned above, our goal was to merely evaluate creative reasoning capabilities. Incorporating heavy domain-specific constraints (which would require complex engineering simulations) would risk confounding reasoning ability with domain knowledge retrieval. By restricting the tasks to self-contained logic puzzles validated by human solvers, we ensure that the ``ground truth'' is defined by the explicit rules of the task, ensuring the results are trustworthy and reproducible.
>
>
> We will include this clarification in the supplementary materials of the paper.
>
> W2) Lack of Human Evaluation: Thank you. To address the feedback regarding evaluation robustness, we conducted a human study with 10 evaluators using the identical prompts employed by the LLM judge. Please refer to our response to Comment W3 from Reviewer vz22 for the detailed analysis. These findings validate the reliability of our LLM evaluation and confirm the strong performance of our Transformational UoT. We hope this addresses the reviewer's main concern.
>
> W3) Positioning of the Paper: We thank the reviewer for the pointers to the related work. We acknowledge that original submission could have covered the literature on reasoning (including, e.g., Alternative Uses tasks) more comprehensively, whereas we mostly focused on the related work we found closest to our work. With the revision, we will significantly expand the related work section to include the referenced line of work and distinguish our work from this line, which admittedly looks closely related at first glance.
>
> However, we wish to clarify the fundamental and significant distinction between our contributions and those of the existing approaches as follows:
>
> - Systematic Creativity vs. Emergent Reasoning: While sophisticated reasoning frameworks like Tree of Thoughts (ToT) or Graph of Thoughts (GoT) enhance problem-solving, their primary objective is typically accuracy or logical consistency. In those frameworks, creative solutions are often an incidental byproduct (or ``luck'') rather than an intended output. In addition, such incidents often happen by borrowing creative reasoning from human user through prompting, rather than via an intrinsic (and intentional) capability. In contrast, our framework is the first to explicitly engineer a mechanism that targets creativity as a primary optimization objective, ensuring novel solutions are generated systematically rather than stochastically.
> - Problem Solving vs. Unconstrained Generation (as in references [1-4] in the reviewer's comments): Unlike standard ``idea generation'' or ``unconventional thinking'' tasks (which often focus on divergent thinking without strict constraints), our setting addresses constrained open-ended problems. We tackle a complex scenario where the output must not only be novel but must also function as a valid solution to a specific problem. This requires balancing creativity with strict logical constraints, a challenge that pure hypothesis generation papers often do not address.
>
>
>
> Q1) Seeds and Thoughts Creation: We would like to clarify the concepts of seeds and thoughts:
>
> - Seeds (Existing Solutions): The baseline solutions ($C$) were created via a human-in-the-loop process. We prompted an LLM for candidates and manually selected standard, functional approaches (e.g., ``Fixed Time Blocks'' for the Bridge task) to ensure they represent accurate but conventional baselines.
> - Thoughts: Unlike the seeds, ``thoughts'' are not pre-defined. They are dynamically generated and decomposed by our UoT process during inference. This allows the model to autonomously explore the search space starting from the manually verified seeds.

---

### Meta-Review · Area_Chair_AwJn · 2026-01-08

**Summary:**

This paper proposes a “Universe of Thoughts” (UoT) framework for creative reasoning in LLMs. Reviewers broadly agree that the topic is timely and the conceptual framing is intuitive. However, several Reviewers also raise concerns that the benchmark and tasks are narrowly constructed and insufficiently validated, and evaluations rely heavily on LLM-as-judge without adequate human verification. Several reviewers also note over-claiming in positioning relative to prior creativity and reasoning literature. Given these concerns, I recommend rejection.

**Reviewer Concerns:**

The rebuttal partially clarifies the high-level motivation and provides additional discussion of the framework’s intent.

**Reviewer Scores:**

I believe Reviewers are unlikely to change their ratings.

---

### Decision · Program_Chairs · 2026-01-26

Reject